# Formation of S- and Z-twist supramolecular micro-ropes by peptide stereoisomers

Hui Yuan[1], Zhongyuan Yang[2], Chengqian Yuan [ID][3], Sudha Shankar[1], Aviad Levin [ID][4], Tiancheng Lv[5], Zihan Wang[2], Wei Sun [ID][5], Jadon Sitton[6], Pierre-Andre Cazade[7], Yoav Dan[8], Yiming Tang [ID][2], Lihi Adler-Abramovich [ID][8], Yi Cao [ID][5], Sigal Rencus-Lazar [ID][1], Damien Thompson [ID][7], Dmitry Kurouski [ID][6], Tuomas P. J. Knowles [ID][4], Linda J. W. Shimon [ID][9], Guanghong Wei [ID][2] ✉, Bin Xue [ID][5] ✉, Rusen Yang [ID][10] ✉ & Ehud Gazit [ID][1] ✉

The intertwined strand arrangement in ropes, from micro- to macro-scale, results in tensile moduli significantly higher than those of single strands. Micro-scale ropes are found in biological systems, most commonly in mechanically-rigid collagen tri-strand arrangements. While human-made macro-ropes possess either left-handed (S) or right-handed (Z) twist, collagen exclusively adopts Z-twist architectures. Despite its natural abundance, the reconstruction and control of these supramolecular ropes in biomimetic systems using minimalist building units remains a fundamental challenge. Here, we demonstrate that cyclo-tryptophan-proline dipeptide stereoisomers self-assemble into complex crystalline supramolecular triple-helical structures. These unique architectures display tunable S- or Z-micro-rope-like twists governed by the configuration of tryptophan residues, as confirmed by co-assembly experiments and molecular dynamics simulations. Tensile testing revealed that these supramolecular micro-ropes exhibit significant moduli. These findings provide a potential platform for designing biomimetic functional helical materials with tunable supramolecular chirality and mechanical strength using minimalist building blocks.

Ropes, dating back to prehistoric Egypt (3500–4000 BC), were initially made by twisting hemp or manila fibers into load-bearing structures composed of intertwined helical strands, featuring either left-handed (S) or right-handed (Z) twists[1,2]. Over time, they have evolved into versatile tools across diverse fields, playing a vital role in the advancement of human civilization[2,3]. Interestingly, rope-like biomolecular structures are ubiquitous in the functional frameworks of plant and animal tissues, ranging from polysaccharides such as cellulose macro-fibers and plant cell walls to polypeptides such as spider silks and collagen fibers[2,4,5].

[1]The Shmunis School of Biomedicine and Cancer Research, George S. Wise Faculty of Life Sciences, Tel Aviv University, Tel Aviv, Israel. [2]Department of Physics, State Key Laboratory of Surface Physics, Key Laboratory for Computational Physical Science (Ministry of Education), Fudan University, Shanghai, People's Republic of China. [3]State Key Laboratory of Biopharmaceutical Preparation and Delivery, Institute of Process Engineering, Chinese Academy of Sciences, Beijing, China. [4]Centre for Misfolding Diseases, Yusuf Hamied Department of Chemistry, University of Cambridge, Cambridge, UK. [5]National Laboratory of State Microstructure, Department of Physics, Nanjing University, Nanjing, Jiangsu, China. [6]Department Biochemistry and Biophysics, Texas A&M University, College Station, TX, USA. [7]Department of Physics, Bernal Institute, University of Limerick, Limerick, Ireland. [8]Department of Oral Biology, The Goldschleger School of Dental Medicine, Gray Faculty of Medical and Health Sciences, Tel Aviv University, Tel Aviv, Israel. [9]Department of Chemical Research Support, Weizmann Institute of Science, Rehovot, Israel. [10]Academy of Advanced Interdisciplinary Research, School of Physics, Xidian University, Xi'an, China. ✉e-mail: ghwei@fudan.edu.cn; xuebinnju@nju.edu.cn; rsyang@xidian.edu.cn; ehudg@post.tau.ac.il

Among these natural materials, collagen, the most abundant protein in mammals and a primary component of the extracellular matrix, adopts a distinctive molecular-level rope-like architecture[6–8]. This structure typically consists of three individual S-polypeptide II helical chains that twist together into a Z-twist superhelical structure stabilized by hydrogen bonds (H-bonds)[9,10]. Each chain comprises a long repeating sequence of Xaa–Yaa–Gly triplets, where the Xaa and Yaa positions are often occupied by proline (Pro, P) and hydroxyproline (Hyp), respectively[11,12]. Owing to its tightly intertwined triple-helical structure, collagen plays a crucial role in providing significant mechanical strength to tissues[13]. Despite its fascinating structures and functions, natural collagen is restricted to a single-handed triple-helical structure due to the stereochemical bias found in nature[14]. Therefore, developing biomimetic materials that emulate and extend the molecular structure of collagen has been the target of both fundamental research as well as applied bioengineering for decades.

Collagen-mimicking peptides, consisting of long repeating triplet units $(X−Y−G)_n$ ($n \geq 6$), have been demonstrated to efficiently reconstruct collagen-like triple-helical conformations[15–19]. The formation of helical structures based on these peptides is favored by special backbone torsion angles, which pose a significant challenge for controlling supramolecular chirality, particularly in heterochiral or mixed-chiral systems. Smaller molecules with fewer chiral centers and reduced conformational complexity also represent promising building blocks, owing to their propensity for forming stable, well-defined assemblies with tunable structures[20–25]. Furthermore, such small entities offer a simpler model for investigating the chirality-dependent assembly of helices, potentially opening opportunities for applications in molecular recognition, chiral sensing, biomedicine, and optoelectronics[26–28]. However, the self-assembly of these short peptides is predominantly based on supramolecular β-sheet secondary structures, largely due to the inherent difficulty in completing hierarchical helical turns using such minimalist molecules, particularly when the goal is to obtain the complex triple-helical crystalline conformation[29,30]. Pro is a key residue in the formation of triple helices[11,12]. Aromatic interactions have been shown to facilitate the self-association process and conformational stabilization and to serve as significant driving forces for assembling diverse supramolecular structures, including helices[31–34]. Moreover, compared to their linear counterparts, cyclic short peptides display increased structural rigidity, greater specificity and binding affinity, as well as resistance to exopeptidases[35]. Inspired by this, we hypothesize that aromatic Pro-based short cyclopeptides are promising candidates for forming stable triple-helical crystals.

Here, we present single-crystal, thermostable, rope-like structures that feature a supramolecular triple-helical arrangement, resulting from the self-assembly of minimalist aromatic cyclic dipeptides composed of tryptophan (Trp, W) and Pro, cyclo-Trp-Pro. In these architectures, dipeptides are linked by H-bonds between the O and N atoms of Trp residues, which govern the molecular twist, thereby forming individual supramolecular helical strands without constraints from backbone torsion angles. Three such strands intertwine to form the supramolecular triple-helical structures, stabilized by interstrand aromatic interactions of Trp residues. The intermolecular binding properties of Trp residues in the structure enable the formation of supramolecular ropes with tunable twists, adopting an S conformation in L-Trp-based assemblies, and a Z conformation in D-Trp-based assemblies, as further elucidated by molecular dynamics (MD) simulations (Fig. 1a). Additionally, a co-assembly strategy was employed to further systematically examine the influence of Trp on the supramolecular structure, confirming that the chirality of Trp directs the packing mode of the resulting co-crystals (Fig. 1a). Owing to their unique supramolecular triple-helical arrangements, these peptide micro-ropes exhibited tensile moduli significantly higher than those of supramolecular single-helical analogues and non-helical crystals.

## Results

### Assembly and structural characterization of cyclo-Trp-Pro dipeptides

Owing to the important role of Pro residues and aromatic interactions in the formation and stability of helical turns[36,37], cyclo-Trp-Pro dipeptides were selected to construct triple-helical-like structures. To investigate conformations with potentially distinct supramolecular chirality, a series of stereoisomers was employed, including the homochiral cyclo-(L)Trp(L)Pro (c-$^L$W$^L$P) and cyclo-(D)Trp(D)Pro (c-$^D$W$^D$P), as well as the heterochiral cyclo-(L)Trp(D)Pro (c-$^L$W$^D$P) and cyclo-(D)Trp(L)Pro (c-$^D$W$^L$P) (Fig. 1a). First, the chemical structures and purity of the cyclic dipeptides were confirmed by analyzing their retention time in high-performance liquid chromatography (HPLC) (Supplementary Fig. 1) as well as via $^1$H nuclear magnetic resonance (NMR) and Fluorine NMR measurements (Supplementary Figs. 2–9). Following a slow cooling crystallization process (see details in "Methods"), the cyclo-dipeptides formed hexagonal prism-like microstructures, with the c-$^L$W$^D$P and c-$^D$W$^L$P assemblies showing larger dimensions at the same concentration (Supplementary Figs. 10–13), as observed using scanning electron microscopy (SEM). The self-assembly pathways of these peptides were further monitored using time-lapse optical microscopy within glass capillaries. All peptides displayed unidirectional growth along the axial dimension and bidirectional variation in the radial dimension (Fig. 1b–e and Supplementary Videos 1–4), consistent with the morphological changes observed using SEM (Supplementary Figs. 10–13). The observed morphological differences are likely influenced by solubility-dependent crystallization kinetics (Supplementary Fig. 14) and by chirality-dependent packing interactions, as supported by the distinct CD spectra of the stereoisomers (Supplementary Fig. 15). Next, to gain deeper insight, the chirality of the assemblies was further characterized by infrared (IR) and vibrational CD (VCD) measurements (Fig. 1f). All assemblies displayed two characteristic peaks in their IR spectra, located approximately at 1667 cm$^{-1}$ and 1632 cm$^{-1}$, corresponding to C=O stretching of the amide I region. However, the dominant feature of the VCD spectra of c-$^L$W$^L$P and c-$^L$W$^D$P was positive VCD couplets with a negative VCD component at higher frequency and a positive VCD component at lower frequency, suggesting an S-shaped supramolecular packing[38,39]. In contrast, c-$^D$W$^L$P and c-$^D$W$^D$P exhibited negative VCD couplets, with a negative VCD component at lower frequency and a positive VCD component at higher frequency, implying a Z-shaped supramolecular packing[38,39].

Powder X-ray diffraction (PXRD) patterns revealed assemblies with high crystallinity that were well aligned with simulated data from single crystals (Fig. 1g and Supplementary Figs. 16–19). Notably, the diffraction peaks of c-$^L$W$^D$P and c-$^D$W$^L$P assemblies differed from those of c-$^L$W$^L$P and c-$^D$W$^D$P assemblies, indicating distinct crystal packings. Such a structural difference appears to be governed by the incorporation of water molecules, as supported by Raman spectroscopy, differential scanning calorimetry (DSC), and thermalgravimetric analysis (TGA) (Fig. 1h–j). Specifically, in c-$^L$W$^D$P and c-$^D$W$^L$P assemblies, a weak peak at ~3574 cm$^{-1}$ was observed in the Raman spectra (Fig. 1h), implying the presence of water molecules in these assemblies[40]. The DSC and TGA measurements further confirmed this finding, revealing a weight loss at 165 °C (defined as $T_l$) for both c-$^L$W$^D$P and c-$^D$W$^L$P crystals, corresponding to the removal of water molecules (Fig. 1i, j). Moreover, c-$^L$W$^L$P, c-$^L$W$^D$P, c-$^D$W$^L$P, and c-$^D$W$^D$P crystals displayed melting temperatures ($T_m$) of 181, 180, 180, and 183 °C, respectively, indicating high structural stability (Fig. 1i). In contrast, natural collagen disassembles at significantly lower temperatures, typically between 10 °C and 40 °C[41], featuring thermal

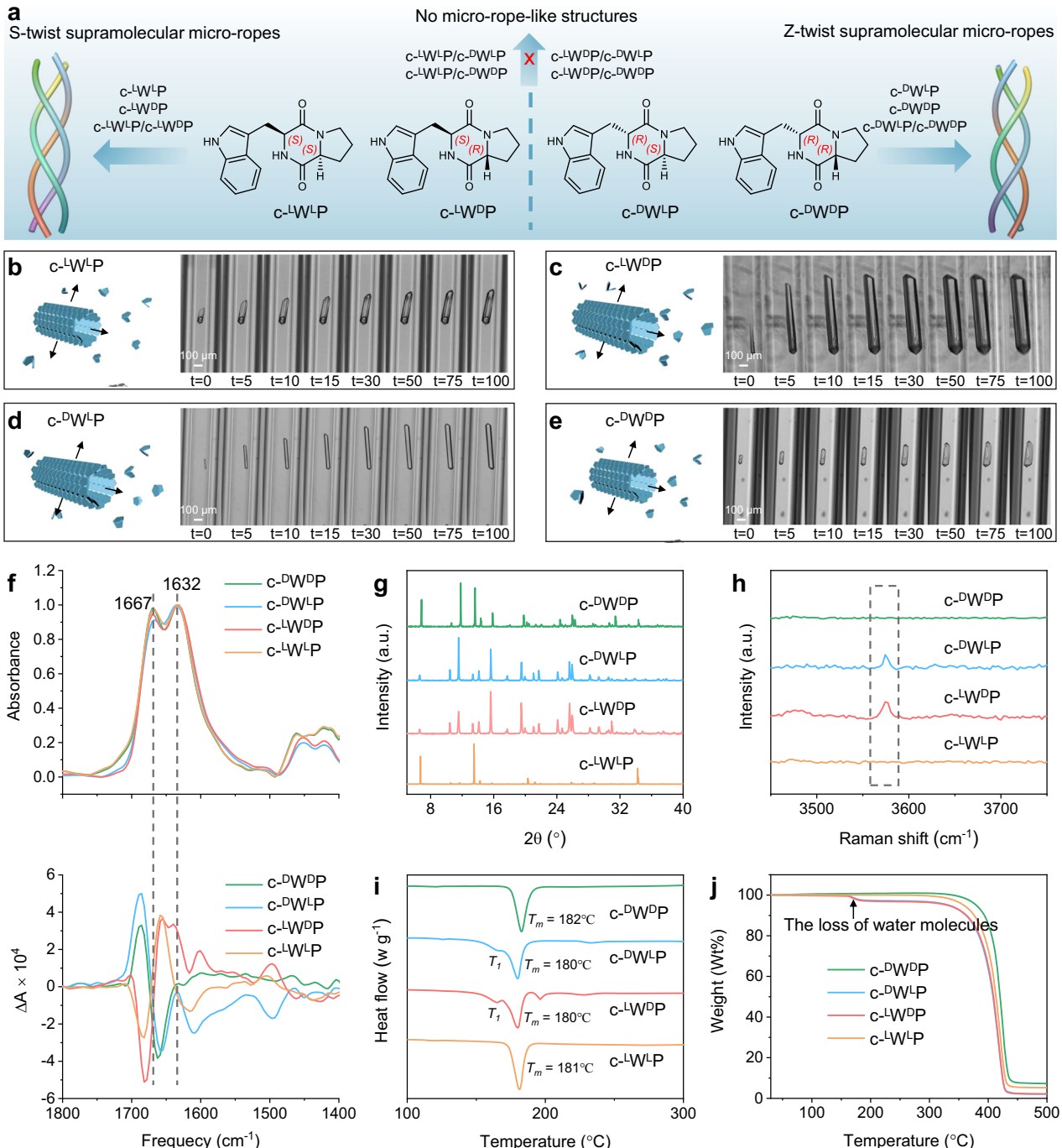

**Fig. 1 | The morphology and structure of cyclo-WP dipeptide assemblies.**
**a** Schematic diagram of the formation of peptide micro-ropes. **b**−**e** Illustration and imaging of the growth of **b** c-LWLP assemblies, **c** c-LWDP assemblies, **d** c-DWLP assemblies, **e** c-DWDP assemblies. Images were taken at 0, 5, 10, 15, 30, 50, 75, 100 min. **f** Top: IR spectra. Bottom: The corresponding VCD. **g** XRD patterns. **h** Raman spectra. **i** DSC curves. **j** TGA curves.

robustness and potential high-temperature applications of the designed peptide assemblies.

**Single-crystal X-ray structure of cyclo-dipeptide stereoisomers**
To understand the supramolecular packing of cyclo-dipeptide stereoisomers, we conducted single-crystal XRD measurements (Fig. 2, Supplementary Figs. 20–23, and Supplementary Tables 1 and 2). The c-LWLP peptide crystallized in the hexagonal space group $P6_3$ (a = 14.9474 Å, b = 14.9474 Å, c = 10.8213 Å), with six molecules per unit cell (Supplementary Fig. 24). Intermolecular H-bonds (NH...C = O, 2.916 Å) between O and N atoms of adjacent diketopiperazine rings in

the Trp segment connected the twistedly arranged molecules, facilitating the formation of a single S-shaped helical strand with six cyclo-dipeptide molecules per repeating unit (Fig. 2a, b). The aromatic rings were aligned along the H-bonds to minimize steric hindrance, thereby optimizing the overall architecture for global energy minimization (Fig. 2b). Notably, three S-helical strands intertwined and extended along the crystallographic $\vec{c}$ direction, forming a supramolecular triple-helical structure with an S conformation (Fig. 2c), featuring a diameter of 15.698 Å and a helical pitch of 35.806 Å. The adjacent helices aligned in parallel, stabilized by intrastrand H-bonds and edge-to-face interactions between side chain indole rings (Fig. 2c and Supplementary

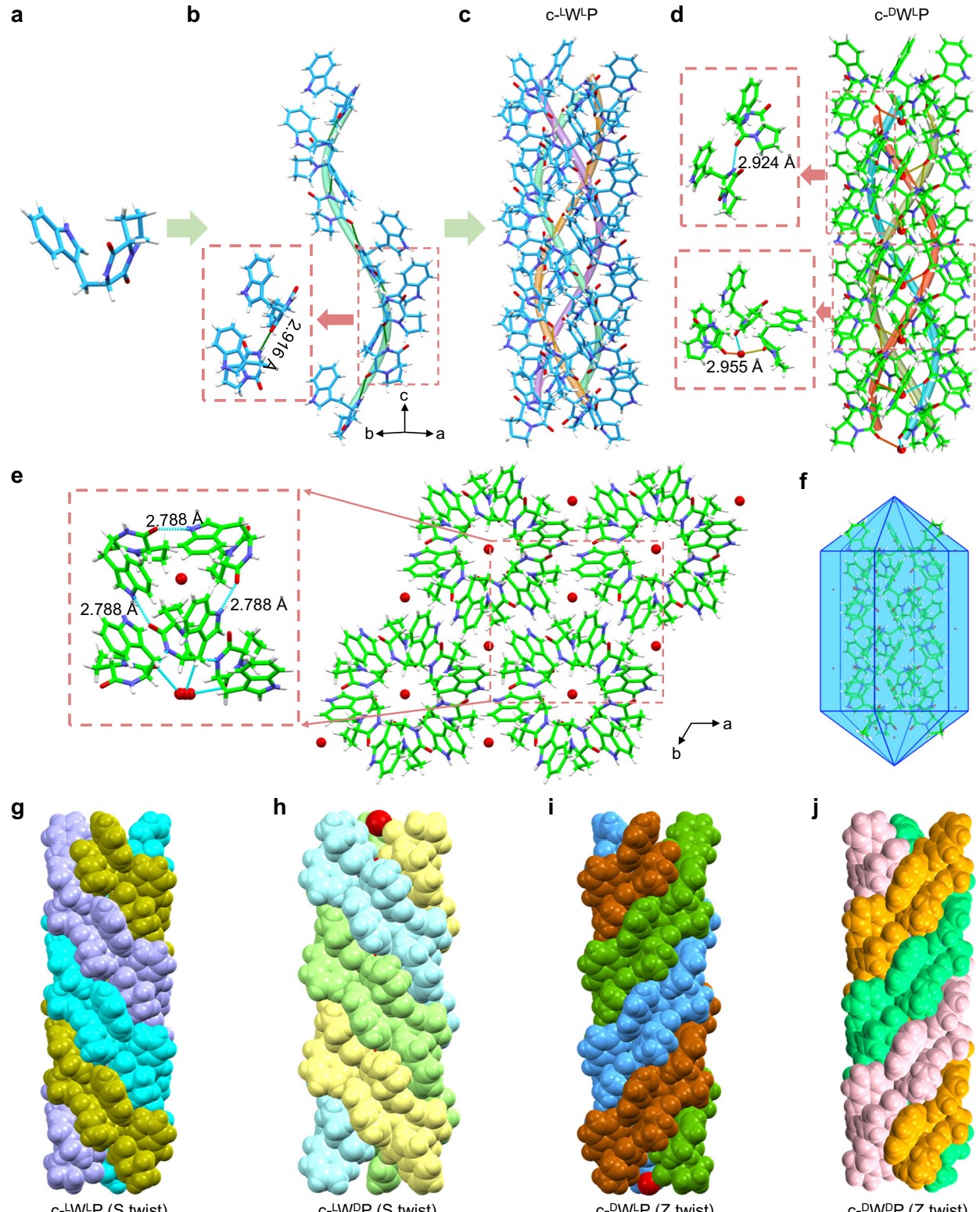

**Fig. 2 | Molecular arrangement of the chiral cyclo-dipeptides showing tunable supramolecular chirality. a–c** Single-crystal structure of c-$^L$W$^L$P. **a** Asymmetric unit. **b** Supramolecular packing into a single-helical strand. **c** Three helical strands twisting together into an S-supramolecular triple-helical structure. **d–f** Single-crystal structure of c-$^D$W$^L$P. **d** Water-mediated Z-supramolecular triple-helical structure. **e** H-bonds connecting the adjacent triple-helical conformations.

**f** Hexagonal prism-like morphology predicted by the BFDH method. Color code: blue, C in c-$^L$W$^L$P; green, C in c-$^D$W$^L$P; periwinkle blue, N; Red, O; White, H. **g–j** Rope-like structures: **g** S-twist peptide rope formed by c-$^L$W$^L$P, **h** Water-mediated, S-twist peptide rope formed by c-$^L$W$^D$P, **i** Water-mediated, Z-twist peptide rope formed by c-$^D$W$^L$P, **j** Z-twist peptide rope formed by c-$^D$W$^D$P. Each helical strand is colored differently to highlight the supramolecular triple-helical structure.

Fig. 25), forming a higher-order micro-rope-like structure. Similar to c-$^L$W$^D$P, the c-$^L$W$^D$P peptide assembly also adopted an S-supramolecular triple-helical structure with a slight increase in the lattice constants ($P6_3$, a = 15.1447 Å, b = 15.1447 Å, c = 10.8688 Å) (Supplementary Fig. 26), owing to the incorporation of water molecules within triple-helical cores and between helices (Supplementary Fig. 27). Additional H-bonds between water molecules and O atoms on diketopiperazine rings, with an OH…C = O distance of 2.959 Å, reinforced the stability of the helical structures. Meanwhile, the water molecules located between helices remained unstructured, not binding to surrounding molecules.

Intriguingly, substituting the $^L$W residue with $^D$W switched the orientation of the supramolecular triple-helical packing, with both c-$^D$W$^L$P and c-$^D$W$^D$P featuring Z-twist micro-rope-like structures (Fig. 2d–f and Supplementary Figs. 28–30). The transformation in structural handedness may be attributed to the Trp residue, which, as described above, forms intrastrand H-bonds connecting adjacent molecules within helical strands and facilitates aromatic interactions that stabilize adjacent triple helices. Despite sharing the same supramolecular chirality, c-$^D$W$^L$P and c-$^D$W$^D$P crystals exhibited slight differences in the molecular arrangement due to the structural water, which was detected in both c-$^D$W$^L$P and c-$^L$W$^D$. In the heterochiral c-WP crystals (c-$^L$W$^D$P and c-$^D$W$^L$P), the diketopiperazine ring exhibits a measurable deviation from planarity, creating additional internal free volume that may accommodate structural water molecules. In contrast, the homochiral crystals maintain an almost planar diketopiperazine ring, consistent with their anhydrous structures. Furthermore, in all cases, these molecules stacked along the helical axis to form hexagonal prism-like predicted morphologies (Fig. 2f and Supplementary Figs. 27e and 30e), based on the Bravais-Friedel-Donnay-Harker (BFDH) theory, closely aligning with the morphologies observed using SEM (Supplementary Figs. 10–13). From the overall supramolecular packing, we noted that both the a and b axes were symmetrical in all the tested crystal structures, while the c axis, considered as the elongation axis of the hexagonal prism, was asymmetrical, thereby leading to bidirectional radial growth and unidirectional axial growth (Fig. 1b–e)[42]. The apparent bidirectional radial growth of c-WP peptides gives rise to the crystals with reduced aspect ratios that are less deformable and more brittle, in contrast to the flexible, rope-like or collagen-fibrillar assemblies. Based on these observations, the engineering of the Trp configuration enables the formation of S-twist micro-rope-like architectures in c-$^L$W$^L$P and c-$^L$W$^D$P peptides, and Z-twist micro-rope-like architectures in c-$^D$W$^L$P and c-$^D$W$^D$P peptides (Fig. 2g–j and Supplementary Videos 5–8). To the best of our knowledge, such triple-helix-like structures from small molecules has rarely been reported[30]. Notably, the building blocks described herein represent the smallest peptides reported to date that are capable of forming rope-like assemblies with tunable structural handedness, paving a new path for the construction of supramolecular triple helices based on the c-WP minimalist building blocks.

## Molecular interactions dictate the chirality of supramolecular triple-helical structures

To elucidate the molecular determinants underlying the single-crystal supramolecular triple-helical structures formed by the c-WP stereoisomers, we performed all-atom MD simulations on 6 × 6 × 6 supercell stacking models of the four chiral combinations of c-WP variants (see "Methods"; Supplementary Fig. 31). We first characterized the interaction patterns among different molecular groups of the c-WP molecules, including the aromatic ($W_{aro}$), NH ($W_{NH}$), side chain ($W_{sc}$), and main chain ($W_{mc}$) of the Trp residue, as well as the main chain ($P_{mc}$) and side chain ($P_{sc}$) of the Pro residue (Fig. 3a). Intermolecular contact probability between each pair of groups (Fig. 3b and Supplementary Fig. 32a) showed that Trp mainchain–mainchain contacts ($W_{mc}$–$W_{mc}$) dominated across all four chiral crystals, underscoring their critical

role in crystal stabilization, followed by $P_{sc}$–$W_{aro}$ and $W_{aro}$–$W_{aro}$ interactions. The $P_{sc}$–$W_{aro}$ and $W_{aro}$–$W_{aro}$ contacts are also significant, likely due to the hydrophobic interactions formed between these groups in different helical bundles (Supplementary Fig. 32b), contributing to the crystal stability. To further identify the specific physical interactions underlying these contact patterns, we calculated the number of H-bonds per unit cell (Fig. 3c). Compared with their homochiral counterparts, the heterochiral crystals (c-$^L$W$^D$P and c-$^D$W$^L$P) exhibited a larger number of H-bonds, implying enhanced structural rigidity and potential stability. The observed increase in H-bonds within the c-$^L$W$^D$P and c-$^D$W$^L$P crystals is primarily attributed to H-bonds formed between structural water molecules and Trp main chains (water-$W_{mc}$) (Supplementary Fig. 33). Statistical analysis of the inner diameters of the triple helices showed that heterochiral crystals exhibited larger inner diameters (Supplementary Fig. 34), which may facilitate the incorporation of water molecules into the triple-helical core and eventually the formation of water-$W_{mc}$ H-bonds in the c-$^L$W$^D$P and c-$^D$W$^L$P crystals.

To further investigate the role of $W_{mc}$–$W_{mc}$ H-bonds in the chirality of the rope-like supramolecular structures, we calculated their rotational angle along the elongation axis of the hexagonal prism (c axis) as a function of the number of crystal layers. Clockwise and counterclockwise rotations corresponded to decreasing and increasing angles, respectively (Fig. 3d–g). In the crystals containing L-Trp residues (c-$^L$W$^L$P and c-$^L$W$^D$P), $W_{mc}$–$W_{mc}$ H-bonds exhibited a clockwise rotation along the elongation direction of the crystals, leading to the formation of S-supramolecular triple-helical structures. Conversely, in the crystals containing D-Trp ($^D$W; c-$^D$W$^L$P and c-$^D$W$^D$P), the H-bonds rotated counterclockwise along the elongation axis, leading to Z-supramolecular triple-helical structures (Fig. 3h, i). These results clearly demonstrate that the chirality of the Trp residue governs the rotational orientation of $W_{mc}$–$W_{mc}$ H-bonds, thereby dictating the macroscopic chirality of the resulting supramolecular triple helix.

Additionally, the role of π−π stacking interactions between the aromatic rings of Trp residues in stabilizing the different chiral c-WP crystals was explored. Consistent with their smaller lattice constants and diameters, the homochiral crystals (c-$^L$W$^L$P and c-$^D$W$^D$P) exhibit slightly enhanced π−π interactions compared to their heterochiral counterparts (Fig. 3j). Based on the spatial arrangements of aromatic rings within the crystal lattices, we classified π−π stacking into three distinct patterns. Patterns I and II refer to inter-helix interactions, while Pattern III refers to intra-helix interactions (Fig. 3k). Free energy landscape analysis revealed that Pattern I corresponds to a tightly connected, low-energy region (Fig. 3i and Supplementary Fig. 35a), with a typical T-shaped π−π stacking. In contrast, the center-of-mass distances between W aromatic rings in Patterns II and III were lower than the maximum separation (≤0.65 nm) for effective π−π stacking. Representative snapshots are shown in Supplementary Fig. 35b. These findings demonstrate that π−π stacking interactions originating from Trp residues play a crucial role in stabilizing the ordered packing of adjacent helices in the supramolecular architectures.

In summary, the H-bond formed between Trp main chains along the elongation direction of the triple helix, together with the hydrophobic stacking interactions between W-W/W-P side chains across different helical bundles, collectively stabilize the triple-helical structure of c-WP crystals. Additionally, the heterochiral crystals (c-$^L$W$^D$P and c-$^D$W$^L$P) form significantly more hydrogen bonds than the homochiral structures (c-$^L$W$^L$P and c-$^D$W$^D$P), primarily due to the involvement of structural water in the H-bond network. This denser hydrogen-bonding network in the heterochiral crystals is expected to result in a higher tensile modulus, which will be examined in the subsequent analysis of the mechanical properties of the c-WP crystals. Moreover, the importance of Trp in forming supramolecular triple-helical structures was further verified by assembling Trp-free, proline-based materials, including benzyloxycarbonyl (O)-protected proline (O-P)

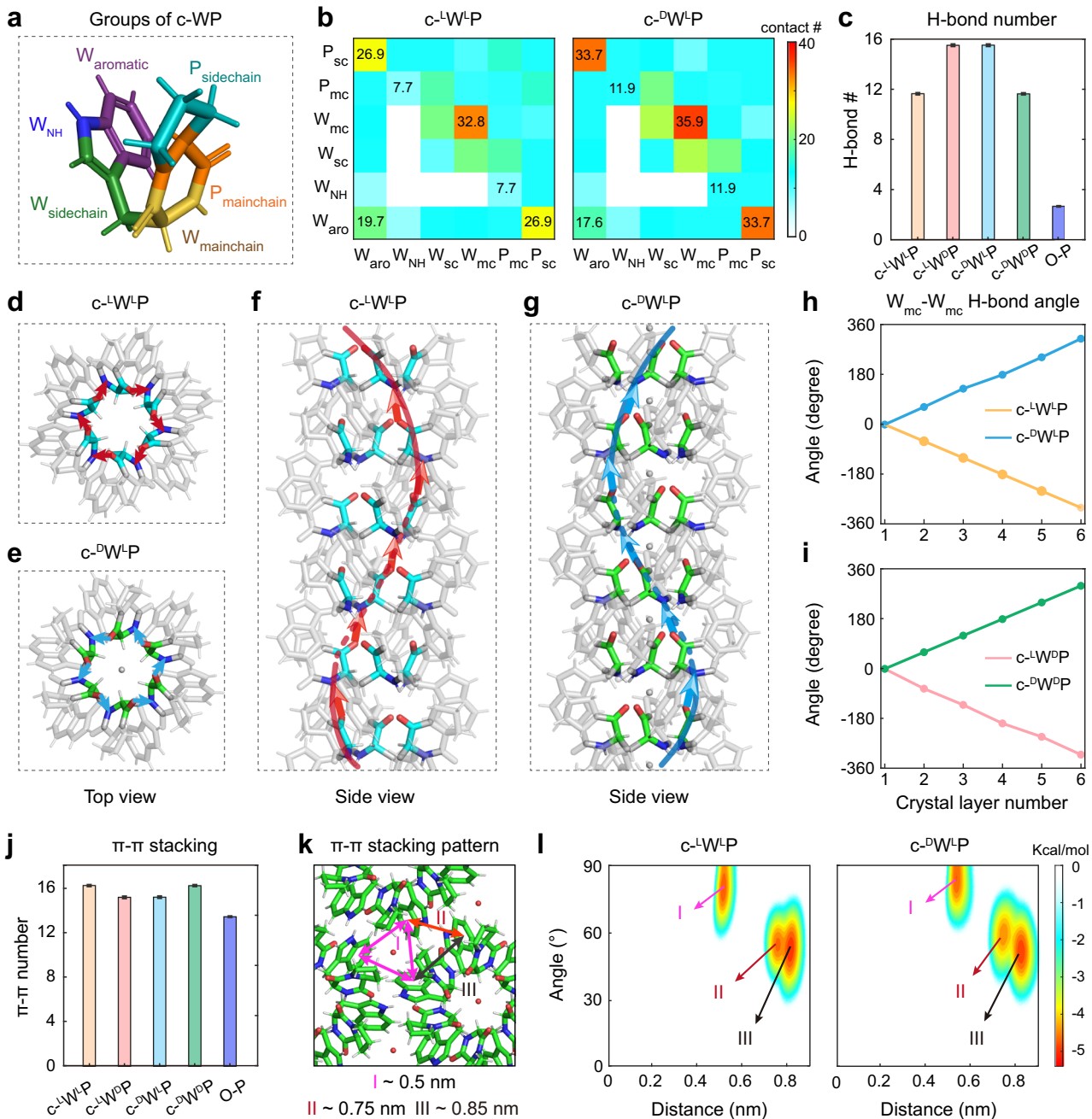

**Fig. 3 | Molecular determinants of single-crystal triple-helical structures formed by the c-WP stereoisomers. a** Different groups of the c-WP molecule, including the aromatic ($W_{aro}$), NH ($W_{NH}$), side chain ($W_{sc}$), and main chain ($W_{mc}$) of the Trp residue, as well as the main chain ($P_{mc}$), and side chain ($P_{sc}$) of the Pro residue. **b** Inter-molecular contact probability maps between different groups of c-WP molecule in the (left) c-$^L$W$^L$P and (right) c-$^D$W$^L$P crystals. The color bar indicates the inter-molecular contact probability. **c** Hydrogen bond number per crystal unit. Error bars represent the standard deviation of the corresponding values over the last 50 ns of the simulations. **d, e** Cross-sectional snapshots of triple helix clusters of **d** c-$^L$W$^L$P and **e** c-$^D$W$^L$P crystals, with the $W_{mc}$–$W_{mc}$ H-bond highlighted in

blue and red, respectively. **f, g** Side view snapshots of the triple helix in **f** c-$^L$W$^L$P and **g** c-$^D$W$^L$P crystal system with $W_{mc}$–$W_{mc}$ H-bond highlighted. **h, i** The rotation angle of $W_{mc}$–$W_{mc}$ H-bonds along the principal crystal axis (*c* axis) as a function of the stacking layer. **j** Number of π–π stacking per crystal unit. Error bars represent the standard deviation of the corresponding values over the last 50 ns of the simulations. **k** Representative snapshots of π–π stacking modes. Patterns I and II depict inter-helix stacking with center of mass distances of -0.5 and -0.75 nm, respectively, while Pattern III represents intra-helix stacking at -0.85 nm. **l** Free energy landscapes of π–π angle versus distance for (right) c-$^L$W$^L$P and (left) c-$^D$W$^L$P crystals.

and cyclo-valine-proline (c-VP), which exclusively formed supramolecular single-helical or non-helical structures (Supplementary Figs. 36–43 and Table 3).

## Co-assembly of cyclo-dipeptide stereoisomers
Deciphering the hierarchical mechanism underlying the transition from single-molecule chirality to supramolecular one is pivotal for

constructing helical assemblies with controllable supramolecular chirality, yet it remains an unsolved challenge[43]. To further explore the relationship between the Trp stereochemistry and supramolecular structures, we employed a co-assembly approach to generate architectures with varied spatial conformations by mixing different configurational cyclo-dipeptides. Utilizing a slow cooling crystallization process, c-$^L$W$^L$P/c-$^L$W$^D$P, c-$^L$W$^L$P/c-$^D$W$^L$P, c-$^L$W$^L$P/c-$^D$W$^D$P, c-$^L$W$^D$P/c-$^D$W$^L$P,

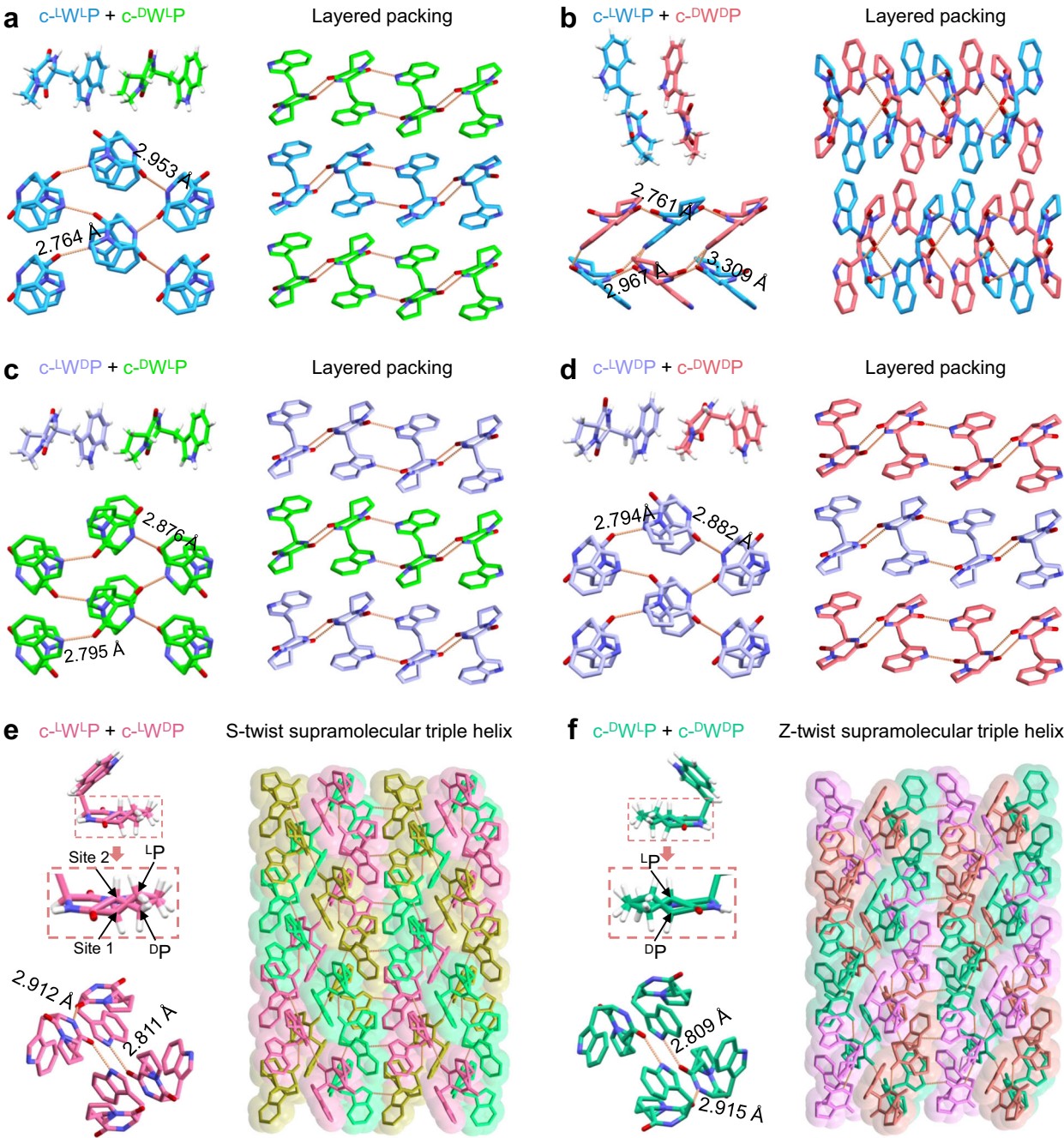

**Fig. 4 | Supramolecular packing of co-crystals to verify the molecular-supramolecular relationship.** The molecule, intrachain H-bonds, and higher-order supramolecular packing of **a** c-$^L$W$^L$P/c-$^D$W$^L$P, **b** c-$^L$W$^L$P/c-$^D$W$^D$P, **c** c-$^L$W$^D$P/c-$^D$W$^L$P, **d** c-$^L$W$^D$P/c-$^D$W$^D$P, **e** c-$^L$W$^L$P/c-$^L$W$^D$P, and **f** c-$^D$W$^L$P/c-$^D$W$^D$P co-crystals. Color code: blue, C in c-$^L$W$^L$P; green, C in c-$^D$W$^L$P; light periwinkle blue, C in c-$^L$W$^D$P; pink, C in c-$^D$W$^D$P; magenta, C in c-$^L$W$^L$P/c-$^L$W$^D$P; shamrock green, C in c-$^D$W$^L$P/c-$^D$W$^D$P; periwinkle blue, N; red, O; white, H. Each helical strand is colored differently to highlight the supramolecular triple-helical structure.

c-$^L$W$^D$P/c-$^D$W$^D$P, and c-$^D$W$^L$P/c-$^D$W$^D$P co-crystals were formed. These co-crystals presented distinct morphologies (Supplementary Figs. 44–49) observed via SEM and different crystal phases (Supplementary Figs. 50–55) determined by XRD, both of which differed from those of individual assemblies. The XRD patterns were consistent with the simulated patterns derived from single-crystal XRD data (Supplementary Figs. 50–55). Additionally, $^1$H NMR spectra confirmed that each co-assembly comprised the two molecules introduced during the growth process (Supplementary Figs. 56–61).

To gain insights into the supramolecular arrangement of the co-assemblies, we performed single-crystal XRD measurements (Supplementary Figs. 62–67 and Supplementary Tables 4–6). The co-assembly

of c-$^L$W$^L$P and c-$^D$W$^L$P yielded a co-crystal with the monoclinic space group $P2_1$ (Fig. 4a and Supplementary Fig. 68), which showed a significant difference in supramolecular packing compared with their individual single-crystal structures (Fig. 2a–e and Supplementary Figs. 24 and 28). In the co-crystal, c-$^L$W$^L$P and c-$^D$W$^L$P formed independent two-dimensional (2D) H-bonded network layers (NH···C=O: 2.953 and 2.764 Å in the c-$^L$W$^L$P layer; 2.883 and 2.798 Å in the c-$^D$W$^L$P layer) aligned along the *ac* plane and stacked in an alternating layered fashion (Fig. 4a). The c-$^L$W$^L$P/c-$^D$W$^D$P co-crystal adopted the orthorhombic *Pbca* space group and featured a mixed-layered structure in which c-$^L$W$^L$P and c-$^D$W$^D$P molecules were integrated within the same H-bonded sheets (Fig. 4b and Supplementary Fig. 69). In the c-$^L$W$^D$P/

c-$^D$W$^L$P co-crystal, peptide monomers stacked into layered architectures composed of alternating c-$^L$W$^D$P and c-$^D$W$^L$P molecular layers, connected by uniform intralayer H-bonds (Fig. 4c and Supplementary Fig. 70), reflecting their opposite molecular chirality. Moreover, the c-$^L$W$^D$P/c-$^D$W$^D$P co-crystal displayed unit cell parameters and supramolecular packing similar to those of the c-$^L$W$^L$P/c-$^D$W$^L$P system, consistent with the mirror symmetry of their chiral compositions (Fig. 4d and Supplementary Fig. 71). The key finding from these co-crystallization experiments is that no triple-helical structures were observed in the co-crystals containing mixed Trp chirality, which disrupts the intrinsic asymmetric spatial stacking of building blocks into the twisted triple-helix motifs.

Next, we sought to analyze co-crystal structures containing Trp residues with consistent chirality. In the c-$^L$W$^L$P/c-$^L$W$^D$P co-crystal, each asymmetric unit contained a single cyclo-dipeptide molecule, with the Trp segment adopting an L-configuration, while the Pro sites were equally occupied by both L-Pro and D-Pro segments (Fig. 4e and Supplementary Fig. 72). Notably, owing to the unchanged configuration of the Trp segment, the c-$^L$W$^L$P/c-$^L$W$^D$P co-crystal retained a supramolecular packing analogous to that of the individual single-crystal structures, displaying an S-supramolecular triple-helical structure (Supplementary Video 9), without being constrained by unmatched backbone torsion angles. Similarly, the c-$^D$W$^L$P/c-$^D$W$^D$P co-crystal, composed of molecules containing D-Trp residues and with the Pro sites equally occupied by both L-Pro and D-Pro, adopted an opposite Z-supramolecular triple-helical structure, connected by intrachain H-bonds and stabilized by aromatic interactions between adjacent chains (Fig. 4f, Supplementary Fig. 73, and Supplementary Video 10). These findings further indicate that supramolecular chirality is predominantly dictated by the stereochemistry of the Trp residue. To our knowledge, these rope-like co-crystals represents the first demonstration that mixed-chirality co-crystals adopt triple helices similar to those of their individual crystals, while previously reported mixed-chirality peptides typically disturb supramolecular helical packing[44–46]. These findings expand the fundamental chiral design principles for constructing triple-helical-like structures with tunable supramolecular chirality based on the minimalist c-WP building blocks.

### Mechanical properties of the cyclo-peptide crystals

Inspired by previously reported collagen assemblies[47], the rope-like structures were anticipated to confer the crystals with significant macroscopic mechanical strength. To examine this hypothesis, the mechanical performance of the crystals at the macroscopic level, including tensile modulus, fracture strain, fracture stress, and work of rupture, was evaluated using standard tensile testing (Fig. 5a–e). As shown in the stress–strain curves (Fig. 5b), the S-twist micro-rope-like peptide crystals exhibited a higher tensile modulus than the Z-twist micro-rope-like peptide crystals, which may stem from variations in hydrogen-bond directions, internal pores, and crystal defects (Supplementary Fig. 74). Notably, water-mediated supramolecular structures demonstrated significantly enhanced mechanical strength (Fig. 5b and d), with c-$^L$W$^D$P crystals reaching a tensile modulus of 0.85 ± 0.28 GPa (Fig. 5c, d). This enhancement can be attributed to the increased number of H-bonds and the additional water-mediated H-bond networks in the triple helix (Fig. 3 and Supplementary Fig. 33), both of which may provide greater resistance to external tensile stress. Consequently, c-$^D$W$^L$P crystals exhibited the second-highest tensile modulus of 0.73 ± 0.15 GPa, followed by c-$^L$W$^L$P crystals (0.40 ± 0.10 GPa) and c-$^D$W$^D$P crystals (0.22 ± 0.08 GPa). Moreover, the tensile moduli of the c-$^L$W$^D$P and c-$^D$W$^D$P crystals were calculated to be ~14 and 3 times higher, respectively, than that of the control O-P crystals (60 ± 10 MPa), which adopt a supramolecular single-helical structure with the decreased number of H-bonds and aromatic interactions calculated by MD simulations (Fig. 3 and Supplementary Fig. 75). Moreover, the water-mediated c-WP crystals exhibited

significantly higher tensile moduli than L-glycine (Gly), L-cysteine (Cys), and L-threonine (Thr) crystals, despite the latter also possessing dense 2D or three-dimensional (3D) H-bond networks but lacking intertwined supramolecular helical structures (Supplementary Figs. 76–79). As summarized in Fig. 5e, the intertwined triple-helical strands, together with the aromatic interactions that seal each strand, play a crucial role in reinforcing the mechanical strength of the peptide micro-ropes.

Next, the compressive Young's modulus of the assemblies along the radial direction at the microscale was determined using nanoindentation based on atomic force microscopy (AFM) (Fig. 5f–h and Supplementary Figs. 80–83). The compressive Young's modulus of the c-$^L$W$^L$P crystal was determined to be 9.5 ± 2.8 GPa, similar to that of the c-$^D$W$^D$P crystal (9.4 ± 2.3 GPa), but higher than those of the c-$^L$W$^D$P (4.6 ± 1.1 GPa) and c-$^D$W$^L$P (4.5 ± 1.7 GPa) crystals (Fig. 5g, h and Supplementary Figs. 80–83). Similarly, the point stiffness of the c-$^L$W$^L$P and c-$^D$W$^D$P crystals was slightly higher than that of the c-$^L$W$^D$P and c-$^D$W$^L$P crystals (Fig. 5g, h and Supplementary Figs. 80–83). The increased Young's moduli and point stiffness in the homochiral crystals may be attributed to higher intrachain H-bonds and slightly stronger π–π interactions along the radial direction (Fig. 3 and Supplementary Fig. 33). The compressive Young's moduli of c-$^L$W$^L$P and c-$^D$W$^D$P crystals are comparable to those of c-$^L$W$^L$P/c-$^L$W$^D$P, c-$^L$W$^L$P/c-$^D$W$^L$P, c-$^L$W$^D$P/c-$^D$W$^L$P, c-$^L$W$^D$P/c-$^D$W$^D$P, and c-$^D$W$^L$P/c-$^D$W$^D$P co-crystal structures, but lower than that of the c-$^L$W$^L$P/c-$^D$W$^P$ co-crystal (Supplementary Figs. 84–89), owing to the dense H-bonded networks and supramolecular packing in the latter. Moreover, the supramolecular triple-helical structures exhibited higher compressive Young's moduli than the control O-P crystals (Supplementary Fig. 90), but lower values than those of control Gly (44 ± 1 GPa), Cys (28.1 ± 1.03 GPa), and Thr (40.95 ± 1.03 GPa) crystals at the microscale (Supplementary Fig. 91)[48,49]. The discrepancy between the macroscopic tensile and nanoscale compressive moduli highlights the dependence of mechanical responses on both loading mode and the testing method in small-molecule crystals.

## Discussion

Here, we demonstrate a set of single-crystal peptide micro-ropes with tunable S- or Z-supramolecular triple-helical structures, which were assembled from minimalistic cyclo-Trp-Pro stereoisomers through H-bonds and aromatic interactions. Supramolecular chirality in these peptides is determined solely by the configuration of a single Trp residue, which modulates intrachain H-bond rotation angles that govern the molecular twist, as revealed by experimental and MD studies. Unlike traditional helical peptides that require specific backbone torsion angles, this single-residue-driven mechanism broadens the design chemical space for triple-helical assembly. Consequently, L-Trp structures (c-$^L$W$^L$P and c-$^L$W$^D$P, and their co-crystal c-$^L$W$^L$P/c-$^L$W$^D$P) exhibited S-twist micro-rope-like conformations, whereas the corresponding D-Trp structures (c-$^D$W$^L$P and c-$^D$W$^D$P and co-crystal c-$^D$W$^L$P/c-$^D$W$^D$P) adopted Z-twist organizations. In contrast, co-crystals with both L- and D-Trp residues (c-$^L$W$^L$P/c-$^D$W$^L$P, c-$^L$W$^L$P/c-$^D$W$^P$, c-$^L$W$^D$P/c-$^D$W$^L$P, and c-$^L$W$^D$P/c-$^D$W$^D$P) displayed non-rope-like conformations resembling previously reported cases, resulting from disruption of spatially asymmetric packing. Moreover, tensile testing revealed that the water-mediated S-twist crystal exhibited an enhanced tensile modulus of 0.85 ± 0.28 GPa. Our work presents a new paradigm of triple-helical materials constructed from minimalist supramolecules, in which single-residue-directed microscale chirality enables precise control over supramolecular chirality even in complex mixed or binary chiral environments. This finding provides a promising platform for exploring chirality-related functionalities, such as enantioselective recognition, and for developing new optical and biomedical materials.

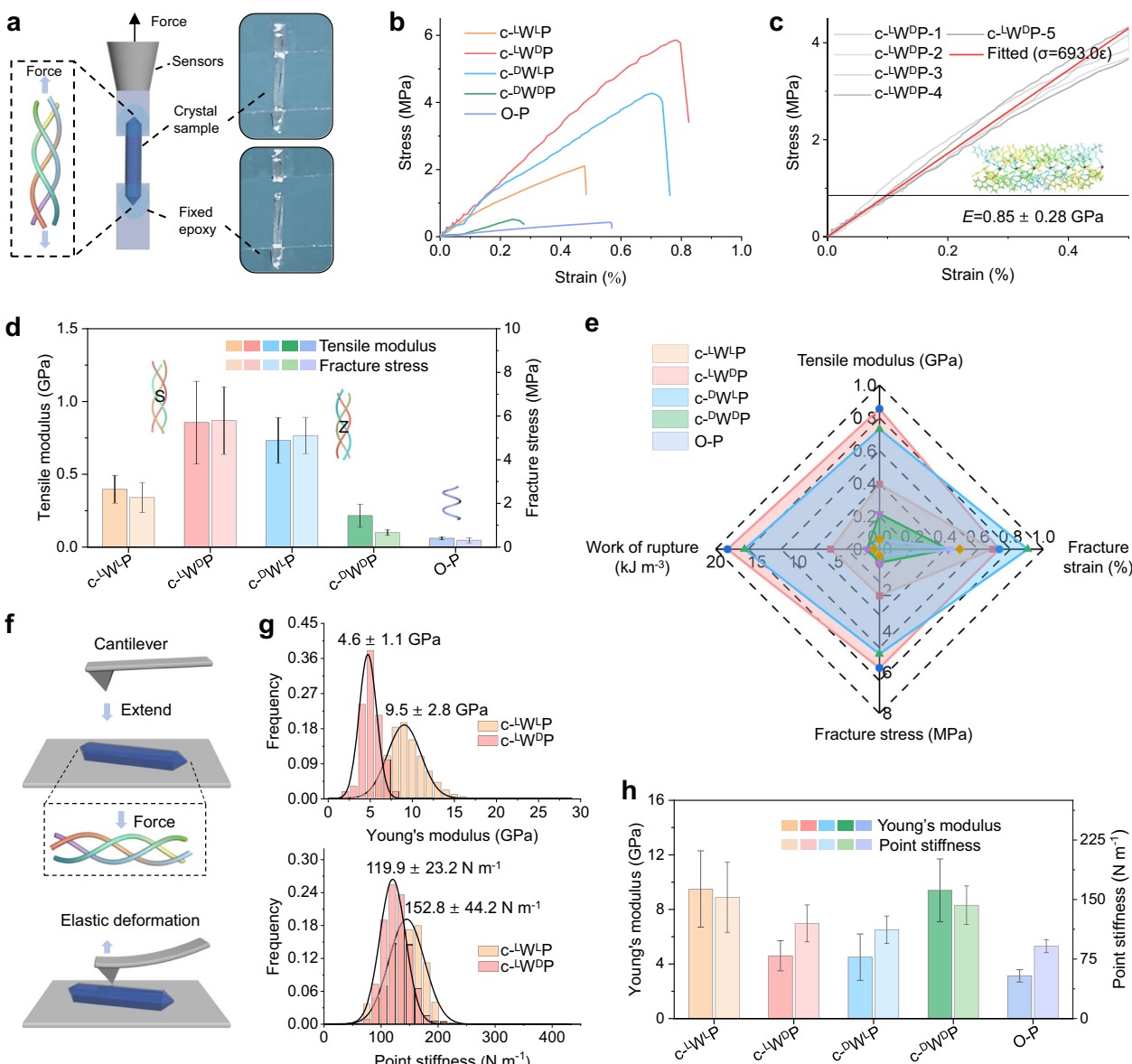

**Fig. 5 | Mechanical properties of the peptide ropes. a** Schematic diagrams of tensile measurements. **b** Typical stress–strain curves of different crystals under tension. **c** Zoom-in on the initial region of typical stress–strain curves of c-$^LW^DP$ crystals. **d** Comparison of tensile modulus and fracture stress obtained from crystals with supramolecular triple-helical and supramolecular single-helical structures. Error bars for the tensile modulus and fracture stress represent the standard deviation from six independent measurements for c-$^LW^LP$, c-$^LW^DP$, c-$^DW^LP$, and c-$^DW^DP$ crystals and from two independent measurements for O-P crystals. **e** Comparison of mechanical parameters of crystals with supramolecular triple-helical and single-helical structures. **f** Schematic diagrams of Young's modulus measurements of single crystals using AFM-based nanoindentation. **g** Statistical Young's modulus distributions and statistical point stiffness distributions of c-$^LW^LP$ and c-$^LW^DP$ determined using AFM-based nanoindentation. **h** Comparison of Young's modulus and point stiffness obtained from nanoindentations for crystals with supramolecular triple-helical and supramolecular single-helical structures. Error bars for Young's modulus represent the standard deviation from five independent measurements.

## Methods

### Self-assembly of peptide crystals
The c-$^LW^LP$, c-$^LW^DP$, c-$^DW^LP$, and c-$^DW^DP$ crystals were formed through a slow cooling crystallization process. Specifically, 2 mg of peptide, purchased from GL Biochem (Shanghai) Ltd., China, was transferred into a 2 mL plastic centrifuge tube. Subsequently, the weighed peptides were dissolved in water at a concentration of 13.3 mg mL$^{-1}$ by heating and shaking at 96 °C. Once the solution became transparent, it was allowed to cool slowly to room temperature, resulting in the formation of clear crystals. The crystals grew to their maximum size after two weeks and were then filtered from the reaction mixture to obtain the crystalline product.

### Co-assembly of peptide co-crystals
The c-$^LW^LP$/c-$^LW^DP$, c-$^LW^LP$/c-$^DW^LP$, c-$^LW^LP$/c-$^DW^DP$, c-$^LW^DP$/c-$^DW^LP$, c-$^LW^DP$/c-$^DW^DP$, and c-$^DW^LP$/c-$^DW^DP$ co-crystals were prepared via a similar slow cooling crystallization process, using different peptide concentrations. For the c-$^LW^LP$/c-$^LW^DP$ co-crystals, c-$^LW^LP$ and c-$^LW^DP$ peptides were individually dissolved in water at a concentration of 14.2 mg mL$^{-1}$ by heating and shaking at 96 °C until clear solutions were obtained. Subsequently, equal volumes of the two solutions were mixed and heated at 96 °C for an additional 0.5 h. The c-$^LW^LP$/c-$^LW^DP$ co-crystal was formed by slowly cooling the solution. Crystals reached their maximum size after two weeks. The c-$^LW^LP$/c-$^DW^LP$, c-$^LW^LP$/c-$^DW^DP$, c-$^LW^DP$/c-$^DW^LP$, c-$^LW^DP$/c-$^DW^DP$, and c-$^DW^LP$/c-$^DW^DP$ co-crystals

were prepared following the same process at concentrations of 16, 10, 10, 16, and 14.2 mg mL$^{-1}$, respectively. These crystalline products were collected by filtration and dried under vacuum.

## Self-assembly of O-P and c-VP crystals

The O-P crystals were obtained by adding 4.5 mL of water into an acetonitrile solution of O-P at a concentration of 400 mg mL$^{-1}$. The O-P crystals reached their maximum size after two weeks and were then collected from the solution by filtration. The c-VP crystals were obtained by slow evaporation. Specifically, 20 mg c-VP was fully dissolved in a methanol-water mixture ($V_{methanol}$:$V_{water}$ = 2:1) at a concentration of 20 mg mL$^{-1}$, and the crystalline products were obtained after one week of evaporation.

## Time-lapse optical microscopy

Upon dissolution of the peptide powders, the solutions were loaded into rectangular glass capillaries (CM Scientific, Silsden, UK), sealed with wax, and imaged using bright-field microscopy. A Nikon Ti-E Inverted fluorescence microscope equipped with a Zyla sCMOS camera was used for time-lapse imaging of c-$^L$W$^L$P, c-$^D$W$^L$P, and c-$^D$W$^D$P assemblies. A custom-built inverted fluorescent microscope (Cairn Scientific, UK) equipped with a Prime BSI sCMOS camera (Teledyne, UK) was used for time-lapse imaging of c-$^L$W$^D$P assemblies. Images and videos were processed using ImageJ software. Time-lapse images were captured at one-minute intervals throughout the growth process, with a total duration of 151 min for c-$^L$W$^L$P, c-$^D$W$^L$P, and c-$^D$W$^D$P, and 105 min for c-$^L$W$^D$P, as shown in Supplementary Videos 1–4.

## Powder X-ray diffraction

Cyclo-Trp-Pro dipeptide crystals or co-crystals were ground into fine pieces using a mortar and then mounted onto a quartz zero-background sample holder. PXRD measurements were conducted using a D8 DISCOVER diffractometer. Data were recorded using Cu Kα radiation over a 2θ range of 5° to 40° at room temperature.

## VCD and IR spectroscopy

VCD and IR spectra were acquired using a ChiralIR-2X spectrometer equipped with MCT detector and DualPEM for enhanced VCD baseline stability. For each measurement, ~20 μl of sample was placed in a BioCell with CaF$_2$ windows and a 6 μm pathlength. During measurements, BioCell was rotated at a constant velocity about the IR beam axis using SyncRoCell (BioTools, Inc.) to eliminate cell and possible sample birefringence. For each sample, VCD and IR spectra were acquired for ~12 h at 8 cm$^{-1}$ spectral resolution. Spectral baselines for VCD and IR were corrected using the acquired VCD and IR spectra of water and water vapor in BioCell under identical conditions. GRAMS/AI 7.0 (Thermo Galactic, Salem, NH) was used for spectral data processing.

## Raman

Raman spectra were acquired using a spectrophotometer (Horiba Jobin Yvon LabRAM HR). The peptide crystals were deposited on a glass slide. A frequency-doubled Nd:YAG laser (λ = 532 nm) was employed as the excitation source. An edge filter was applied to suppress the Rayleigh line. The scattered light was collected using a thermoelectrically cooled CCD detector (Synapse, operating at −70 °C). Spectral dispersion was achieved using a diffraction grating with 600 grooves per millimeter.

## TGA and DSC

Thermogravimetric and heat flow curves were obtained for all samples using a TGA/DSC 3$^+$ series instrument (Mettler Toledo, Switzerland). Baseline adjustment was performed prior to testing using two empty platinum crucibles with loose covers. Samples were heated from room temperature at a rate of 10 K min$^{-1}$ to the target temperature using the same instrument.

## Processing and structural refinement of crystal data

High-quality cyclo-Trp-Pro dipeptide crystals or co-crystals of suitable size were covered with Paratone oil (Hampton Research), placed on a Mateen cryo-loop, and flash-frozen in liquid nitrogen. A Rigaku Synergy-R system equipped with a HyPix-Arc150 detector with Cu Kα radiation was used to collect the crystal and co-crystal data at 120 K. Data for the c-$^D$W$^L$P, c-$^D$W$^D$P, and c-$^L$W$^D$P/c-$^D$W$^D$P crystal and co-crystal were collected using the Rigaku Synergy S system with PILATUS 300 K detector under Mo Kα radiation. The c-$^L$W$^D$P data was analyzed on a Rigaku 007HF Rigaku XtaLAB P200 diffractometer with Cu Kα radiation at 113.15 K. The O-P data were collected on a Rigaku Synergy-R system equipped with an XtaLAB Synergy-R detector, using Cu Kα radiation at 293 K.

Diffraction data were processed via the Rigaku CrysAlis Pro software, and the crystal structures were solved and refined using Bruker SHELXTL. Non-hydrogen atoms were positioned in calculated positions and refined in the riding model. Details of the data collection and refinement parameters are summarized in Supplementary Tables 1–6, while the final CIF files are available in the Supplementary Information. Crystallographic data of c-$^L$W$^L$P, c-$^L$W$^D$P, c-$^D$W$^L$P, and c-$^D$W$^D$P have been deposited in the CCDC under deposition numbers 2465098-2465101. The crystal structures of the c-$^L$W$^L$P/c-$^L$W$^D$P, c-$^L$W$^L$P/c-$^D$W$^L$P, c-$^L$W$^L$P/c-$^D$W$^D$P, c-$^L$W$^D$P/c-$^D$W$^L$P, c-$^L$W$^D$P/c-$^D$W$^D$P, and c-$^D$W$^L$P/c-$^D$W$^D$P co-crystals are available under deposition numbers 2465116-2465121. The crystal structure of O-P and C-VP assemblies was found in deposition number 2465132 and 2504998.

## Simulation systems

Four distinct chiral combinations of c-WP (c-$^L$W$^L$P, c-$^L$W$^D$P, c-$^D$W$^L$P, and c-$^D$W$^D$P) molecules and the O-P molecules were used in the simulations. The bonded and van der Waals parameters of c-WPs were taken from the Amber force field. The restrained electrostatic potential charges were obtained by fitting to quantum-mechanical-calculated electrostatic potentials using the Ambertools package[50]. The initial structures of c-WP and O-P were taken from the single-crystal structures solved through the single-crystal XRD experiments described herein. For each simulation system, one 100 ns MD simulation was conducted as shown in Supplementary Fig. 27. The simulation cell dimensions were set to hold 6 × 6 × 6-unit cells, with the box parameters adjusted according to the crystallographic properties of each chiral crystal (Supplementary Fig. 27a, b). The RMSD to the first frame and the average contact number per unit cell showed that the equilibrium has been reached (Supplementary Fig. 27c, d). Periodic boundary conditions were applied along all three directions to reproduce bulk-phase properties. The parameters of the monoclinic box are listed in Supplementary Table 7.

## MD simulations

All MD simulations were performed using the GROMACS 2022.6 package[51]. Explicit water was modelled using the TIP3P water model[52]. Electrostatic interactions were calculated using the Particle Mesh Ewald method[53] with a real space cut-off of 1.4 nm. The same cut-off was used for the calculation of van der Waals interactions. A 1 fs integration time step was employed, enabled by constraining bond lengths via the LINCS algorithm[54] for c-WP molecules and the SETTLE method[55] for water molecules. Prior to production runs, energy minimization was performed using the steepest descent algorithm. Temperature coupling was implemented separately for c-WP molecules and solvent using a velocity-rescaling thermostat, maintaining a constant temperature of 300 K[56]. Final production simulations were carried out in the NVT ensemble without any restraints.

## Data analysis

Analyses of the simulation data were carried out using in-house scripts and the tools implemented in the GROMACS package. The statistical analyses were performed using the data generated in the last 50 ns (i.e., 50–100 ns) of each trajectory. The all-atom root-mean-square deviation (RMSD) was calculated with reference to the initial frame. A H-bond was considered to be formed when the distance between H-bond donor (D) and acceptor (A) was ≤ 0.35 nm and the D-H···A angle was ≥ 150°. A contact between two non-hydrogen atoms was defined when they were within 0.54 nm (for carbon-carbon pairs) or 0.46 nm (for any other atom pairs). The contact number between two groups was defined as to the number of atom pairs satisfying the contact criterion. The H-bond angle was determined by calculating the angle between the projection of the H-bond in the xy-plane and the projection of the reference H-bond (the H-bonds in the first crystal layer). Two aromatic rings were considered to form a π−π stacking interaction when their centroid distance fell within 0.65 nm[31]. The free energy landscape of aromatic stacking interactions was calculated by the formula, $-RT\ln P$ (distance, angle), where $P$ (distance, angle) was the probability of two rings with a certain centroid distance and angle. The inner diameter was determined by calculating the diameter of the circle formed by connecting the $C_\alpha$ atoms of three Trp residues within the same layer. Graphical analysis and structure visualization were performed using the Pymol software[57].

## Tensile modulus

Tensile stress–strain measurements were performed using a universal testing machine (5944, Instron, USA) equipped with a 10 N load cell at room temperature (23 °C) under ambient air conditions. Single-crystal samples exceeding 5 mm in length were prepared for tensile testing. The diameters of the crystals were measured using an optical microscope (IX73, Olympus, Japan) to determine the cross-sectional area. Each crystal was mounted by fixing both ends to parallel glass holders using epoxy glue, and the holders were secured to the tensile testing machine. After the glue was fully cured, the samples were stretched to fracture at a constant speed of 0.3 mm min⁻¹. During the test, tensile force and displacement were continuously recorded. Stress was calculated by dividing the applied force by the cross-sectional area of the crystal. The Young's modulus was determined based on the initial linear region of the stress–strain curve within a strain range of 0–0.2%. The fracture stress was defined as the maximum stress reached during stretching. c-ᴸWᴸP, c-ᴸWᴰP, c-ᴰWᴸP, and c-ᴰWᴰP crystals with lengths >5 mm were selected for testing, while their co-crystals were excluded because their lengths (<2 mm) were inadequate for tensile measurements. Therefore, O-P crystals with a supramolecular single-helical structure, as well as Gly, Cys, and Thr crystals with non-helical structures, were employed as controls.

## AFM-based nanoindentation

AFM-based nanoindentation experiments were conducted using a commercial AFM system (Nanowizard IV, JPK, Germany). Crystals were dispersed onto freshly cleaved mica substrates and gently blown with nitrogen to remove loosely attached particles. During each measurement, the AFM cantilever was positioned on the crystal surface, and a 10 × 10 μm area was scanned in tapping mode to locate a flat region. Nanoindentation was then performed on a 5 × 5 μm area within the flat region in Quantitative Imaging (QI) mode under the following conditions: resolution of 256 × 256 pixels, Z range of 0.05 μm, approach and retraction speed of 30 μm s⁻¹, Z resolution of 80,000 Hz, and a maximum loading force of 800 nN.

RTESPA-525 cantilevers (Bruker, nominal tip radius ~10 nm, spring constant ~200 N m⁻¹, pyramidal tip half-angle θ < 10°) were used for all measurements. The cantilever was extended toward and retracted from the sample surface, and force–displacement curves were recorded. The Young's modulus ($E$) of the crystal was calculated by fitting the approach curve using the Hertz model:

$$F = \frac{4}{3}\frac{E}{(1-v^2)}\sqrt{R}\delta^{\frac{3}{2}} \tag{1}$$

where $F$ is the applied force, $\delta$ is the indentation depth, $R$ is the tip radius, and $v$ is the Poisson's ratio (0.3). Point stiffness was determined as the ratio of normal force to the sample deformation, with correction for cantilever deflection based on the force-displacement curves. For statistical robustness, nanoindentation was carried out at more than 6 distinct regions, with 1–2 flat areas randomly selected within each region. To minimize tip-related bias, at least 3 different cantilevers were employed. All data were analyzed and 2D maps were reconstructed using JPK Data Processing software (version 7.0.46, JPK Instruments).

## Data availability

All data are available from the corresponding author upon request. Crystallographic data for the structures reported in this Article have been deposited at the Cambridge Crystallographic Data Centre, under deposition numbers CCDC 2465098-2465101, 2465116-2465121, 2465132, and 2504998. These data can be obtained free of charge via https://www.ccdc.cam.ac.uk/structures/. Source data are provided with this paper.

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

## Acknowledgements

This work was supported in part by the ISF—Israel Science Foundation by Grant no. 3246/23 within the China-Israel Cooperative Scientific Research (Grant No. 52361145848), Twin2pipsa—Twinning for excellence in biophysics of protein interactions and self-assembly grant (101079147). G.W. and B.X. acknowledge financial support from the National Natural Science Foundation of China (Grant Nos. 12374208 and T2322010) and the Natural Science Foundation of Shanghai (Grant No. 22ZR1406800). R.Y. thanks Natural Science Foundation of Shaanxi Province (Grant No. 2020JCW-15). P.-A.C. and D.T. thank Science Foundation Ireland (SFI) for financial support under Grant Number 12/RC/2275_P2 (SSPC). L.A.-A. acknowledges the support of the European Research Council (ERC), under the European Union's Horizon 2020 research and innovation program (grant agreement no. 948102), and the

Israel Science Foundation (grants no. 2422/24). We wish to thank Dr. Davide Levy for PXRD measurements, Dr. Evelina Nikeshparg for assisting with the Raman measurements. Lastly, we are grateful to all members of the Gazit group for their insightful discussions and valuable input.

## Author contributions

H.Y., E.G., and R.Y. developed the concept of S- and Z-twist supramolecular micro-ropes by peptide stereoisomers. H.Y. conducted the peptide crystallization, materials characterization, and data analysis. S.S. carried out the solubility, HPLC, and NMR measurements. A.L., Y.D., T.P.J.K., and L.A.-A. performed time-lapse optical microscopy measurement and analyzed the data. C.Y. crystallized the O-P amino acid and conducted TGA and DSC measurements. J.S. and D.K. conducted the IR and VCD measurements. L.J.W.S. performed the single-crystal diffraction measurement and solved the crystal structures. Z.Y., Z.W., Y.T., and G.W. performed the all-atom single-crystal molecular dynamics and drafted the simulation part of the manuscript. B.X., T.L., W.S., and Y.C. measured the tensile modulus and Young's modulus and analyzed the data. H.Y. drafted the manuscript, while E.G., R.Y., B.X., G.W., Y.C., D.T., S.R.-L., T.P.J.K., D.K., L.A., Y.C., and P.A.C. revised it. All authors provided feedback on the manuscript.

## Competing interests

The authors declare no competing interests.
