## [Transparent Peer Review file · Nature Communications]

Formation of S- and Z-twist supramolecular micro-ropes by peptide stereoisomers

Corresponding Author: Professor Ehud Gazit

Version 0:

Reviewer comments:

Reviewer #1

(Remarks to the Author)

The authors have addressed my previous comments and suggested in a comprehensive and clearly structured manner.

Reviewer #2

(Remarks to the Author)

The revisions have mostly addressed my previous concerns.

In my opinion, this manuscript is now suitable for publication in this journal.

Attached is my detailed feedback with a few minor suggestions.

[Editorial Note: See end of file]

Reviewer #3

(Remarks to the Author)

Version 2:

Reviewer Comments

Reviewer #1 (Remarks to the Author):

The authors have addressed my previous comments and suggested in a comprehensive and clearly structured manner.

Response:

We thank the reviewer for the positive assessment and for acknowledging the improvements made to the revised manuscript.

Reviewer #2 (Remarks to the Author):

The revisions have mostly addressed my previous concerns.

In my opinion, this manuscript is now suitable for publication in this journal.

Attached is my detailed feedback with a few minor suggestions.

Response: We are heartened by the reviewer's positive assessment that our manuscript is now suitable for publication. We sincerely appreciate the constructive feedback provided throughout the revision process, which has greatly enhanced the impact and clarity of our work.

Comment 2: OK. These new data addressed my concerns. However, since these are crucial data for the main negative control of the study (e.g., Supplementary Fig. 76), the reviewer highly recommends including the data in the main Figure, e.g., in Fig. 3 (c and j).

We are grateful for the reviewer's positive feedback and the insightful suggestion regarding the data presentation.

Response: We are grateful for the reviewer's positive feedback and the insightful suggestion regarding the data presentation. We agree that these negative control data are fundamental to the study's conclusions. Following the reviewer's recommendation, we have moved the data from Supplementary Figure S76 to the revised Figure 3 (now panels c and j), and the detail is shown below. We believe this adjustment has significantly improved the clarity and impact of

Fig. 3: Molecular determinants of single-crystal triple-helical structures formed by the c-WP stereoisomers.

(a) Different groups of the c-WP molecule, including the aromatic (W_{aro}), NH (W_{NH}), side chain (W_{sc}), and main chain (W_{mc}) of the Trp residue, as well as the main chain (P_{mc}), and side chain (P_{sc}) of the Pro residue. (b) Inter-molecular contact probability maps between different groups of c-WP molecule in the (left) $c\text{-}^{\text{L}}\text{W}^{\text{L}}\text{P}$ and (right) $c\text{-}^{\text{D}}\text{W}^{\text{L}}\text{P}$ crystals. The color bar indicates the inter-molecular contact probability. (c) Hydrogen bond number per crystal unit. Error bars represent the standard deviation of the corresponding values over the last 50 ns of the simulations. (d-e) Cross-sectional snapshots of triple helix clusters of (d) $c\text{-}^{\text{L}}\text{W}^{\text{L}}\text{P}$ and (e) $c\text{-}^{\text{D}}\text{W}^{\text{L}}\text{P}$ crystals, with the W_{mc} -

W_{mc} H-bond highlighted in blue and red respectively. (f-g) Side view snapshots of the triple helix in (f) $c^L W^L P$ and (g) $c^D W^L P$ crystal system with $W_{mc}-W_{mc}$ H-bond highlighted. (h-i) The rotation angle of $W_{mc}-W_{mc}$ H-bonds along the principal crystal axis (c axis) as a function of the stacking layer. (j) Number of $\pi-\pi$ stacking per crystal unit. Error bars represent the standard deviation of the corresponding values over the last 50 ns of the simulations. (k) Representative snapshots of $\pi-\pi$ stacking modes. Patterns I and II depict inter-helix stacking with center of mass distances of ~ 0.5 and ~ 0.75 nm, respectively, while Pattern III represents intra-helix stacking at ~ 0.85 nm. (l) Free energy landscapes of $\pi-\pi$ angle versus distance for (right) $c^L W^L P$ and (left) $c^D W^L P$ crystals.

Comment 3: OK. But please specific in the figure description which tendon from the rabbits were included in this study described in Figure R1. Are they Achilles tendons, patellar tendons, or rotator cuff tendons, etc.? The anatomical variations of tendons, such as whether the tendon is weight-bearing, energy-storing, or positional, directly affect its collagen fiber alignment, cross sectional area, and proteoglycan content, leading to distinct tensile moduli and viscoelastic behaviors.

Response: We thank the reviewer for this insightful comment. We agree that the anatomical source and functional specialization of tendons (e.g., weight-bearing vs. positional) significantly influence their collagen hierarchy and, consequently, their biomechanical properties.

In this study, rabbit patellar tendons were used for the experiments described in Figure R1. As a major component of the extensor mechanism, the patellar tendon is a weight-bearing and positional tissue characterized by highly aligned Type I collagen fibers and a significant cross-sectional area, which are adapted to withstand high tensile loads.

Please find the reviewer's feedback to the revision in dark green under each comment response.

Reviewer 2:

In this manuscript, the authors describe the self-assembly of minimalist aromatic cyclic dipeptides, cyclo-(L/D)Trp-(L/D)Pro, resulting in single-crystal, thermostable, rope-like structures that feature a unique supramolecular triple-helical arrangement. The authors further demonstrate that these supramolecular micro-ropes exhibit moduli comparable to those of natural collagen fibers. This study shows considerable originality in the field of mimicking collagen molecular assembly. Previous literature on synthetic collagen-mimetic assemblies has almost exclusively focused on triple-helical peptides using (Gly-X-Y)_n (n=5-10) as structural units. To the best of this reviewer's knowledge, there are no precedents employing dipeptides as small as two amino acids. The study provides detailed, thorough, and multifaceted characterizations of several assembled structures. The majority of the data are of high quality and are presented clearly.

However, this reviewer has significant concerns regarding the validation of the work's core conclusions/correlation, which may strongly impact the manuscript's suitability for publication in this journal.

Response: We thank the referee for the positive and encouraging comments.

Comment 1: The study repeatedly emphasizes that the supramolecular triple-helical structure is crucial for the stability and mechanical properties of the assemblies. Interestingly, however, although all four cWP crystals in Fig. 5d possess a triple-helical assembly (Fig. 2) and exhibit nearly identical numbers of hydrogen bonds and pi-pi stacking interactions per crystal unit (Fig. 3c & 3j), why do their tensile moduli differ so significantly (Fig. 5d)? The

manuscript does not appear to provide a clear explanation for this substantial structure-property discrepancy. The marked difference in their tensile moduli seems to suggest that the common triple-helical supramolecular assembly might not be the specific structural foundation responsible for their mechanical properties.

Response to Comment 1: We thank the reviewer for raising this important point. Structural factors that contribute to the observed differences in the tensile moduli are outlined below.

First, the helical structure is stabilized by both H-bonding and aromatic interactions (Fig. 3), whereas the tensile modulus of the crystals is governed primarily by the intermolecular H-bonding network aligned along the loading direction. Notable differences are observed in both the number and organization of hydrogen bonds. As shown in Fig. 3, the heterochiral crystals (LD and DL) exhibit an approximately 33% increase in the total number of H-bonds compared with the homochiral structures (LL and DD) (Fig. 3c). In the heterochiral crystals, the additional inter-peptide hydrogen bonds are oriented along the elongation direction of the triple helix. Moreover, water molecules in the heterochiral crystals form extensive hydrogen bonds with the main-chain oxygen atoms of the Trp residues (Supplementary Fig. 35), giving rise to a denser hydrogen-bonding network that enhances intermolecular connectivity and strengthens their mechanical properties. Together, these results demonstrate that both the increased number of hydrogen bonds and the participation of water molecules in the triple-helix H-bonding network contribute to the higher tensile moduli observed for the LD and DL assemblies.

Second, within the anhydrous or anhydrated pair, the left- and right-handed micro-ropes possess comparable numbers of hydrogen bonds and π - π interactions, yet still show a difference in tensile modulus. This distinction does not arise from the interaction count *per se*, but may stem from subtle chirality-dependent variations in packing geometry, such as small differences in hydrogen-bond directions, which may influence the efficiency of load transfer along the triple-helical packing.

In addition, as observed with many supramolecular structures and crystals, internal voids and crystal defects may also influence the measured mechanical properties [*Nano Lett.* **6**, 616-621 (2006)]. c-WP crystals feature various pore structures (Supplementary Fig. 74), which lead

to lower measured tensile moduli than the theoretical values. While these effects cannot be fully quantified with the current data, they may contribute to the remaining discrepancies that cannot be explained solely by the supramolecular packing geometry.

Taken together, these points suggest that the mechanical response of the c-WP crystals results from the combination of (i) differences in the total interaction network (hydrated vs. anhydrous), (ii) chirality-dependent packing geometry (S vs. Z), and (iii) unavoidable microstructural factors such as pores or defects. We have modified this structure-property relationship in the revised manuscript (page 15, lines 310-318, page 19, lines 403-406 and page 19, lines 397-401) and Supplementary Figure 74. The details are shown below.

In summary, the H-bond formed between Trp main chains along the elongation direction of the triple helix, together with the hydrophobic stacking interactions between W-W/W-P side chains across different helical bundles, collectively stabilize the triple-helical structure of c-WP crystals. Additionally, the heterochiral crystals (c-^LW^DP and c-^DW^LP) form significantly more hydrogen bonds than the homochiral structures (c-^LW^LP and c-^DW^DP), primarily due to the involvement of structural water in the H-bond network. This denser hydrogen-bonding network in the heterochiral crystals is expected to result in a higher tensile modulus, which will be examined in the subsequent analysis of the mechanical properties of the c-WP crystals.

This enhancement can be attributed to the increased number of H-bonds and the additional water-mediated H-bond networks in the triple helix (Fig. 3 and Supplementary Fig. 33), both of which may provide greater resistance to external tensile stress.

As shown in the stress–strain curves (Fig. 5b), the S-twist micro-rope-like peptide crystals exhibited a higher tensile modulus than the Z-twist micro-rope-like peptide crystals, which may stem from variations in hydrogen-bond directions, internal pores and crystal defects (Supplementary Fig. 74).

Supplementary Figure 74: SEM images of (a) $c\text{-LWP}$, (b) $c\text{-LWDP}$, (c) $c\text{-DWP}$, and (d) $c\text{-DWDP}$ assemblies showing pore structures.

Reviewer response: OK

Comment 2: Concurrently, complete data for the key negative control, the single-helical O-P assembly, seem to be missing. For instance, this reviewer was unable to locate the number of hydrogen bonds and pi-pi interactions per crystal unit for this structure in either the main text or the Supplementary Information. The absence of these data makes it difficult to properly assess the correlation between the triple-helical structure and the mechanical properties of the assemblies.

Response to Comment 2: We thank the reviewer for this constructive comment. Following this valuable suggestion, we have performed additional MD simulations on the single-helical O-P assembly and calculated the number of hydrogen bonds and π - π stacking interactions per crystal unit. This allows us to make a direct comparison with the c-WP triple-helical assemblies (see below and in Figs. 75 and 76 of the revised Supplementary Information). Our calculation shows that the number of hydrogen bonds in the O-P single-helical crystal is significantly lower than that in the c-WP triple-helical crystal and the number of π - π stacking interactions is also reduced, albeit to a lesser extent. The markedly weakened hydrogen-bonding interactions in the O-P single-helical crystal would lead to the inferior tensile modulus and mechanical performance compared to the WP triple-helical assembly. These results further support our conclusion that the triple-helical architecture is critical for achieving the superior

tensile properties of the micro-rope structures.

We have included these simulation results of the O-P crystal and discussed the correlation between supramolecular architecture and tensile properties in the revised main text (page 19, lines 408-412) and Supplementary Figures 75 and 76. The details are shown below:

Moreover, the tensile moduli of the c -^LW^{DP} and c -^DW^{DP} crystals were calculated to be ~ 14 and 3 times higher, respectively, than that of the control O-P crystals (60 ± 10 MPa), which adopt a supramolecular single-helical structure with the decreased number of H-bonds and aromatic interactions calculated by MD simulations (Supplementary Figs. 75 and 76).

Supplementary Figure 75: The molecular packing and structural stability of the O-P crystal. (a-b) The initial conformation of MD simulations in two different views of a $10 \times 10 \times 6$ supercell crystal structure: (a) top view and (b) side view. The detailed molecular packings in the crystal structure are highlighted by enlarged views. (c-d) Time evolution of (c) all-atom RMSD of O-P molecules and (d) inter-molecular contact number of the crystal. The RMSD and the inter-molecular contact numbers of each system remain stable over the 100 ns simulation time, confirming the stability of the constructed crystal structures.

Supplementary Figure 76: Comparison of interactions in the triple-helical WP crystal and the single-helical O-P crystal. (a) Number of hydrogen bonds per unit cell. (b) Number of π - π stacking per unit cell.

Reviewer response: OK. These new data addressed my concerns. However, since these are crucial data for the main negative control of the study (e.g., Supplementary Fig. 76), the reviewer highly recommends including the data in the main Figure, e.g., in Fig. 3 (c and j).

Comment 3: The measurement of a material's tensile modulus can be influenced by numerous conditional factors. Rather than solely citing literature values (lines 379-381), a more persuasive approach might involve measuring tendon tissue or collagen fibers under the same conditions for direct comparison with the crystalline materials described here. Furthermore, given that natural collagen fibers—which are based on long-chain biomolecules with intermolecular covalent crosslinks—are structurally quite distinct from the supramolecular crystals described, a more comprehensive evaluation of the mechanical strength exhibited by these cWPs would be beneficial. To properly contextualize their performance among similar materials, the crystals prepared in this work should also be compared (experimentally or via literature data) to other molecular crystals based on hydrogen bonds or pi-pi interactions—such as ice (hydrogen bonds between water molecules), other dipeptides, or perhaps DNA crystals.

Response to Comment 3: We thank the reviewer for the constructive comment. Following this valuable suggestion, we have measured the tensile modulus of rabbit tendon tissues under the same testing conditions used for the c-WP crystals (see Figure R1). The rabbit tendon tissues exhibited a tensile modulus of 0.21 ± 0.08 GPa, which is significantly lower than that of the water-mediated S-micro-ropes (0.85 ± 0.28 GPa).

As the reviewer noted, the hierarchical architecture of natural collagen fibers, composed

of long-chain biomolecules with covalent intermolecular crosslinks, differs from those of the supramolecular crystals explored in this work. Therefore, we removed the comparison between c-WP crystals and collagen, as their mechanical behaviors are not directly comparison.

To more appropriately contextualize the mechanical performance of the c-WPs within the broader category of small-molecule crystals, we have included comparisons to amino acid crystals, including L-glycine (Gly), L-cysteine (Cys), and L-threonine (Thr). These materials are classical minimalist molecular crystals stabilized predominantly by hydrogen bonds and therefore may serve as suitable controls. Under comparable tensile testing conditions, the Gly, Cys, and Thr crystals exhibited tensile moduli of 0.09 ± 0.03 GPa, 0.19 ± 0.05 GPa, and 0.21 ± 0.07 GPa, respectively, all of which are lower than those of c-WP crystals, especially the water-mediated c-WP crystals (0.85 ± 0.28 GPa for c-^LW^DP crystals) (Supplementary Figs. 77-80). Notably, previous reports showed high compressive Young's moduli for Gly (44 ± 1 GPa) and Thr crystals (40.95 ± 1.03 GPa) based on nanoindentation [*Angew. Chem. Int. Ed.* **54**, 13566-13570 (2015); *Nat. Commun.* **12**, 1326 (2021)]. Moreover, the compressive Young's modulus of the Cys crystals was measured to be 28.1 ± 8.2 GPa based on AFM nanoindentation (Supplementary Fig. 92), which is also higher than those of the c-WP crystals. These findings highlight the dependence of mechanical responses on both loading mode and testing method in small-molecule crystals. This contrast underscores the important role of the supramolecular triple-helical packing in enhancing the tensile modulus of the c-WP crystals. The corresponding comparative experiments and analyses have been added to the revised manuscript (page 19, lines 413-416 and page 20, lines 435-441) and Supplementary Figures 77-80 and 92, as detailed below.

Moreover, the water-mediated c-WP crystals exhibited significantly higher tensile moduli than L-glycine (Gly), L-cysteine (Cys), and L-threonine (Thr) crystals, despite the latter also possessing dense 2D or three-dimensional (3D) H-bond networks but lacking intertwined supramolecular helical structures (Supplementary Figs. 77-80).

Moreover, the supramolecular triple-helical structures exhibited higher compressive Young's moduli than the control O-P crystals (Supplementary Fig. 91), but lower values than those of control Gly (44 ± 1 GPa), Cys (28.1 ± 1.03 GPa), and Thr (40.95 ± 1.03 GPa) crystals

at the microscale (Supplementary Fig. 92)^{48,49}. The discrepancy between the macroscopic tensile and nanoscale compressive moduli highlights the dependence of mechanical responses on both loading mode and the testing method in small-molecule crystals.

Supplementary Figure 77: (a) Photograph of a Gly self-assembled crystal. (b) Powder X-ray diffraction pattern showing that the Gly assemblies belong to the α -Gly structure. (c,d) Gly molecules packed into a layered supramolecular structure. CCDC ref. no. 193596¹.

Supplementary Figure 78: (a) Photograph of a Cys self-assembled crystal. (b) Powder X-ray diffraction pattern of Cys assemblies. (c,d) Cys molecules interconnected by 3D hydrogen-bond

networks. CCDC ref. no. 683376².

Supplementary Figure 79: (a) Photograph of a Thr self-assembled crystal. (b) Powder X-ray diffraction pattern of Thr assemblies. (c,d) Thr molecules interconnected by dense 3D hydrogen-bond networks. CCDC ref. no. 2024959³.

Supplementary Figure 80: (a) Typical stress-strain curves of Gly, Cys, and Thr crystals under tension. (b) Comparison of tensile moduli obtained from Gly, Cys, and Thr crystals.

Supplementary Figure 92: (a) Topographic Young's modulus map, (b) topographic point stiffness map, (c) statistical Young's modulus distribution, and (d) statistical point stiffness distribution of Cys. Scale bar = 1 μm .

References:

- 48 Azuri, I. et al. Unusually large Young's moduli of amino acid molecular crystals. *Angew. Chem. Int. Ed.* **54**, 13566-13570 (2015).
- 49 Karothu, D. P. et al. Mechanically robust amino acid crystals as fiber-optic transducers and wide bandpass filters for optical communication in the near-infrared. *Nat. Commun.* **12**, 1326 (2021).

Supplementary references:

- 1 Langan, P., Mason, S. A., Myles, D. & Schoenborn, B. P. Structural characterization of crystals of α -glycine during anomalous electrical behaviour. *J. Appl. Crystallogr.* **58**, 728-733 (2002).
- 2 Kolesov, B. A., Minkov, V. S., Boldyreva, E. V. & Drebushchak, T. N. Phase transitions

in the crystals of L- and DL-cysteine on cooling: intermolecular hydrogen-bond distortions and side-chain motions of thiol groups. 1. L-cysteine. *J. Phys. Chem. B* **112**, 12827-12839 (2008).

3 Karothu, D. P. et al. Mechanically robust amino acid crystals as fiber-optic transducers and wide bandpass filters for optical communication in the near-infrared. *Nat. Commun.* **12**, 1326 (2021).

Figure R1: Typical stress-strain curves and summarized average tensile modulus of rabbit tendon tissue. The tendon samples were obtained from New Zealand White rabbits in accordance with the regulations and guidelines of the Ethics Committee of Drum Tower Hospital affiliated with the Medical School of Nanjing University and in compliance with the Institutional Animal Care and Use Committee (IACUC) requirements (approval number: IACUC-D2310006).

Reviewer response: OK. But please specific in the figure description which tendon from the rabbits were included in this study described in Figure R1. Are they Achilles tendons, patellar tendons, or rotator cuff tendons, etc.? The anatomical variations of tendons, such as whether the tendon is weight-bearing, energy-storing, or positional, directly affect its collagen fiber alignment, cross-sectional area, and proteoglycan content, leading to distinct tensile moduli and viscoelastic behaviors.

Comment 4: Minor: Although the structural units of the triple-helical peptide-based systems described by the following literature are significantly larger than the dipeptide system described here, they might still be worth mentioning as relevant background:

<https://pubmed.ncbi.nlm.nih.gov/40021488/>

<https://pubmed.ncbi.nlm.nih.gov/40344344/>

<https://pubmed.ncbi.nlm.nih.gov/27410188/>

Response to Comment 4: We thank the reviewer for the helpful comments. We have cited the relevant references in the revised manuscript (page 24, lines 515-521), as details belows.

References:

17 Jin, P., Rafizadeh, D. N., Zhao, H. & Chenoweth, D. M. β -Turn mimicking crosslinking provides hyperstability and fast folding kinetics for short collagen triple helices. *ChemBioChem* **26**, e202400834 (2025).

18 Fiala, T. *et al.* Hyperstable, Minimal-length, and blunt-ended collagen heterotrimers. *Angew. Chem.*, e202503353 (2025).

19 Zhang, Y., Herling, M. & Chenoweth, D. M. General solution for stabilizing triplehelical collagen. *J. Am. Chem. Soc.* **138**, 9751-9754 (2016).

Reviewer response: OK

Comment 5: Minor: Line 147: The label for Fig 1j is missing in the figure description.

Response to Comment 5: We thank the reviewer for the helpful comments. We have corrected the writing mistake in the revised manuscript (page 7, line 146). The revised figure description is highlighted below.

Fig. 1: The morphology and structure of cyclo-WP dipeptide assemblies.

(j) TGA curves.

Reviewer response: OK